# Hyperbaric oxygen therapy improves symptoms, brain's microstructure and functionality in veterans with treatment resistant post-traumatic stress disorder: A prospective, randomized, controlled trial

**Keren Doenyas-Barak**[1,2]* , **Merav Catalogna**[1], **Ilan Kutz**[1], **Gabriela Levi**[1], **Amir Hadanny**[1,2], **Sigal Tal**[2,3], **Shir Daphna-Tekoha**[4,5,6], **Efrat Sasson**[1], **Yarden Shechter**[1], **Shai Efrati**[1,2,7]

1 The Sagol Center for Hyperbaric Medicine and Research at Shamir Medical Center Zerifin, Zerifin, Israel, 2 Sackler School of Medicine Tel-Aviv University, Tel-Aviv, Israel, 3 Radiology Department, Shamir Medical Center, Zerifin, Israel, 4 Kaplan Medical Center, Rehovot, Israel, 5 Faculty of Social-Work, Ashkelon Academic College, Ashkelon, Israel, 6 Emili Sagol Creative Arts Therapies Research Center, Mount Carmel, Haifa, Israel, 7 Sagol School of Neuroscience, Tel-Aviv University, Tel-Aviv, Israel

☯ These authors contributed equally to this work.
* Kerendoenyas@gmail.com

## Abstract

### Introduction

Post-traumatic stress disorder (PTSD) is characterized by changes in both brain activity and microstructural integrity. Cumulative evidence demonstrates that hyperbaric oxygen therapy (HBOT) induces neuroplasticity and case-series studies indicate its potentially positive effects on PTSD. The aim of the study was to evaluate HBOT's effect in veterans with treatment resistant PTSD.

### Methods

Veterans with treatment resistant PTSD were 1:1 randomized to HBOT or control groups. All other brain pathologies served as exclusion criteria. Outcome measures included clinician-administered PTSD scale-V (CAPS-V) questionnaires, brief symptom inventory (BSI), BECK depression inventory (BDI), brain microstructural integrity evaluated by MRI diffuse tensor imaging sequence (DTI), and brain function was evaluated by an n-back task using functional MRI (fMRI). The treatment group underwent sixty daily hyperbaric sessions. No interventions were performed in the control group.

### Results

Thirty-five veterans were randomized to HBOT (N = 18) or control (n = 17) and 29 completed the protocol. Following HBOT, there was a significant improvement in CAPS-V scores and no change in the control (F = 30.57, P<0.0001, Net effect size = 1.64). Significant improvements were also demonstrated in BSI and BDI scores (F = 5.72, P = 0.024 Net effect size =

**Data Availability Statement:** All relevant data are within the manuscript and its Supporting Information files.

**Funding:** The study was funded by the Research Fund of Shamir Medical Center, Israel.

**Competing interests:** AH and ES works for AVIV scientific. SE is a shareholder in AVIV scientific.

0.89, and F = 7.65, P = 0.01, Net effect size = 1.03). Improved brain activity was seen in fMRI in the left dorsolateral prefrontal, middle temporal gyri, both thalami, left hippocampus and left insula. The DTI showed significant increases in fractional anisotropy in the fronto-limbic white-matter, genu of the corpus callosum and fornix.

## Conclusions

HBOT improved symptoms, brain microstructure and functionality in veterans with treatment resistant PTSD.

## Introduction

Post-traumatic stress disorder (PTSD) is a complex, chronic, and debilitating psychiatric disorder that develops in response to severe psychological traumatic exposure. PTSD is characterized by intrusive thoughts, nightmares and flashbacks of past traumatic events, avoidance of trauma reminders, hypervigilance, sleep disturbances and persisting dysregulation of the stress response [1]. These protracted symptoms lead to considerable social, occupational, and interpersonal dysfunctions. The global cross-national lifetime prevalence of PTSD reported by the World Health Organization (WHO) is 3.9% [2], while among combatants the prevalence can be as high as 30% [3]. Unfortunately, the current available treatments, including medications and trauma focused psychotherapy, have limited effect and nearly half of the patients suffer from treatment resistant PTSD [4].

New brain imaging techniques enable better understanding of the pathophysiology responsible for developing PTSD. It is now clear that traumatic events cause long-term changes of brain activity and microstructural integrity. The dominant trauma-related pathologies are demonstrated in the frontal-limbic circuit, amygdala, hippocampus and prefrontal cortex [5–8].

Hyperbaric oxygen therapy (HBOT) includes the inhalation of 100% oxygen at pressures exceeding 1 atmosphere absolute (ATA), thus enhancing the amount of oxygen dissolved in the body tissues. Many of the beneficial effects of HBOT can be explained by improvements in tissue/cerebral oxygenation. However, it is currently understood that the combined action of intermittent hyperoxia and hyperbaric pressure, triggers both oxygen and pressure sensitive genes [9]. Additionally, increases in cerebral metabolic rates, restoration of mitochondrial functions, stimulation of cell proliferation and maturation of endogenous neural stem cells, and induction of anti-inflammatory, angiogenic and neurogenic factors have all been demonstrated after HBOT(9). Cumulative evidence from post-stroke and traumatic brain injury (TBI) studies demonstrate that HBOT induces neuroplasticity in the chronic metabolic dysfunctional brain regions even years after the brain insult [10,11]. Recent studies have also demonstrated HBOT induced neuroplasticity and significant clinical improvements in patients with fibromyalgia, including those in whom fibromyalgia was induced by child abuse [12,13].

The potential beneficial effects of HBOT on PTSD were investigated in combat veterans with TBI which is commonly combined with PTSD. In most of the studies, a significant clinical improvement in PTSD symptoms was demonstrated [14–20]. However, to the best of our knowledge, none of these studies focused on PTSD as a stand-alone pathology.

The aim of this study was to evaluate the effect of HBOT on clinical outcomes, brain functionality and brain microstructural integrity in veterans suffering from treatment resistant combat associated PTSD.

## Materials and methods

### Study design and patients

The study was a randomized, prospective controlled trial conducted at the Sagol Center for Hyperbaric Medicine and Research at the Shamir Medical Center, Israel, between March 2018 and October 2019. The protocol was approved by the Shamir Institutional Review Board (199/17) and registered in the National Institute of Health Clinical Trials Registry (NCT03466554).

Patients were referred to the study by their psychiatrist or psychotherapist, or applied for the study after reading an advertisement in their veterans' social media groups. The study included male veterans, age 25 to 60 years old, with combat associated, treatment resistant PTSD lasting at least four years prior to their inclusion. Patients were recruited if they had persistent residual debilitating PTSD symptoms, were exposed to at least one trauma focused therapy and pharmacotherapy, and fulfilled the CAPS questionnaire diagnostic criteria for PTSD. Exclusion criteria included history of TBI or any other brain pathology, active malignancy, substance use at baseline (except for prescribed cannabis, and only if nebulized or taken as a tincture), current manic or psychotic episodes, serious current suicidal ideation, severe or unstable physical disorders or major cognitive deficits at baseline, HBOT for any reason prior to study enrollment, chest pathology incompatible with pressure changes (including active asthma), ear or sinus pathologies incompatible with pressure changes, inability to perform an awake brain MRI and active smoking.

Cognitive evaluation at baseline was performed using the computerized cognitive testing battery "Neurotrax". Cognitive scores are presented as normalized scores according to age and education groups, on an IQ-style scale, where 100 is the mean normalized score and one standard deviation equals to 15 points [21].

### Randomization and masking

Included participants were 1:1 randomly assigned to the treatment or control group according to a computer-generated randomization list. Assessors were blinded to the participants' allocation.

### Procedures

After receiving detailed information regarding study procedure and signing an informed consent form, participants underwent a baseline evaluation which included a review of their medical history, a physical examination, a psychological interview by two senior clinicians, questionnaires and brain imaging. HBOT was given in addition to the patients' pre-inclusion psychotherapy. Participants in the control group continued with their pre-inclusion psychotherapy program and did not receive any hyperbaric treatment. No additional psychotherapy or trauma focused therapy was given as part of the study protocol.

Participants were evaluated at baseline and after three months of HBOT or control.

HBOT: Participants were treated in a multiplace chamber (HAUX-Life-Support GmbH) for a total of 60 daily sessions, five days a week. Each session consisted of 90 minutes exposure to 100% oxygen at 2 ATA with five-minute air breaks every 20 minutes.

Participants in both treatment and control groups continued their psychological and pharmacological treatments as they did before their inclusion. Any changes in the frequency of psychological treatments or pharmacotherapy doses were reported and documented. Monthly meetings with study investigators were scheduled during both treatment and control periods. Unscheduled visits were provided as needed.

## Outcomes

The primary objective was defined as the change in the clinician-administered DSM-V (CAPS-V) PTSD scale score from baseline. The brief symptom inventory–18 (BSI-18), and Beck depression inventory-II (BDI-II) questionnaires served as secondary clinical endpoints. Changes in brain MRI diffuse tensor imagine (DTI) sequence and n-back task in functional MRI (fMRI) were also analyzed as secondary endpoints.

*CAPS-V* is a structured interview-based test that consists of 30 items. Items are rated on a 0 to 4 severity scale. Twenty of the items reflect the severity of DSM-V PTSD symptoms and served as the primary endpoint. The score ranges between 0 and 80, with higher scores indicating more severe PTSD symptoms. The interview was administered by a study investigator, under the supervision of the study psychiatrist at baseline and 1 to 4 weeks after the end of the HBOT or control period.

In addition, participants completed the following questionnaires at baseline and 1 to 4 weeks after the end of the study period:

*Beck depression inventory II (BDI-II)*—BDI-II is a widely used psychometric tests for measuring the severity of depression. It consists of 21 multiple-choice questions and a self-report inventory about how the subject has been feeling in the last week. Each answer is scored on a scale value of 0 to 3. The scored ranges between 0 and 63, with higher scores indicating more severe depression symptoms.

*The brief symptom inventory–18 (BSI-18)*—The BSI-18 contains 18 items in three symptom scales: somatization (6 items), depression (6 items), and anxiety (6 items). Each item is rated on the same 0 to 4 scale that reflects symptom severity in the last seven days, and the sum of all responses yields a global severity index (GSI). Scores range between 0 and 72, with the higher scores indicating worse symptoms.

**Imaging data acquisition.** MRI scans were performed on a MAGNETOM Skyra 3T Scanner, configured with a 20-channel receiver head coil (Siemens Healthcare, Erlangen, Germany). Functional imaging data consisted of 128 volume measurements of gradient-echo (EPI) blood oxygen level dependent (BOLD) contrast sequences. Scan parameters: TR = 3000 ms, TE = 30 ms, flip angle = 90˚, voxel size = 3.0 x 3.0 x 3.0 mm, distant factor = 25%, FOV = 192 mm$^2$, within slice resolutions of 64×64, and 36 contiguous slices parallel to the AP-PC plane. Diffusion whole brain images were acquired with the following parameters: 63 axial slices, slice thickness = 2.2 mm, voxel size = 1.8 x 1.8 mm, TR = 10,300 ms, TE = 89 ms, and matrix = 128 x 128 mm. Diffusion gradients were applied along 30 noncollinear directions (b = 1000 s/mm$^2$) and one volume without diffusion weighting. T1-weighted images were acquired with 3D MPRAGE sequences in sagittal orientation with 0.9 mm isotropic resolution. Sequence parameters: TR = 2,000 ms, TE = 2.41 ms, flip angle = 8˚, TI = 928 ms, FOV = 245 x 245, and 192 contiguous slices.

**Functional task design.** The N-back working memory task is one of the most popular paradigms for functional neuroimaging studies, which refers to temporary storage and manipulation of information. In this study, we used a block design paradigm, consisting of two-condition alternating blocks (0-Back and 2-Back) over a course of eight cycles. Each block consisted of a series of 12 letters. Each letter was presented for 1500 ms, followed by a 1500 ms fixation interval. During the 0-Back condition, participants were asked to respond by pressing a button (ResponseGrip, NordicNeuroLab Inc., Norway) when a target Hebrew letter "ג" was presented. In the 2-Back condition, participants were asked to respond when the current letter was identical to the one presented two trials back. The ratio of target to non-target letters presented in each block is 3/4:12. Participants rehearsed a practice version of the test with a technician outside the scanner to ensure comprehension of the task demands. NordicAktiva,

(NordicNeuroLab Inc., Norway, www.nordicneurolab.no) was used for stimuli presentation, performance accuracy, and response time acquisition.

**MRI data analysis.** Preprocessing of the raw diffusion data, and calculation of DTI-FA (fractional anisotropy) maps were performed using ExploreDTI, and included corrections for eddy current distortion and participant motion. Spatial normalization was performed for each patient based on the mean diffusion image using the ICBM template, based on T1 contrast. The normalization parameters were applied to the DTI maps. Finally, spatial smoothing with a kernel size of 6 mm full width half maximum (FWHM) was applied. To avoid partial volume bias in the statistical map, and to limit statistical testing to white matter, FA maps were thresholded at 0.2.

Analysis of the time series BOLD data was performed using statistical parametric mapping software SPM12 (Welcome Department of Cognitive Neurology, Institute of Neurology, University College London, UK), through a standard preprocessing procedure. All images were initially slice-time corrected, realigned and resliced using a 6-parameter rigid body spatial transformation to correct head motion, and normalized to the MNI space (Montreal Neurological Institute) by using the unified segmentation normalization algorithm. Finally, spatial smoothing was performed using a 6 mm FWHM Gaussian kernel. The general linear model were applied on a subject level. The design matrix incorporated the task and the six spatial axes movement repressors. The task repressors were modeled as a boxcar function, and were convolved with a canonical hemodynamic response function. A high-pass filter (cutoff of 128 s) was applied to account for slow signal drift. All parametric maps thresholds were set at $P < 0.05$ false discovery rate (FDR) corrected for multiple comparisons. The mean percent BOLD signal change was calculated within spherical regions of interest (6 mm radius of gray matter volume), obtained from this analysis, and centered at the peak t value coordinates.

## Statistical analysis

**Sample size.** Since there was no previous data from prospective studies on the potential beneficial effects of HBOT on PTSD, we followed the recommendations of Hertzog [22] for a sample size determination. A small to medium effect size of 0.3 in a repeated measures ANOVA design, with a power of 85% and an alpha of 5%, a total of 28 participants would be required. Adding a 15% dropout rate would require 32 patients in total.

**Data analysis.** Unless otherwise stated, continuous data were expressed as means ± standard-deviations. Independent and dependent t-tests with a two-tail distribution were performed to compare variables between and within the two groups, when a normality assumption held according to the Kolmogorov-Smirnov test. Net effect sizes were evaluated using Cohen's d method, defined as the improvement from baseline after HBOT minus control three months improvement divided by the pooled standard deviation (SD) of the composite score. Positive effect sizes indicate improvement. Categorical data were expressed in numbers and percentages and compared by chi-square/Fisher's exact test to identify significant variables. A value of $P < 0.05$ was considered significant. Continuous parameters correlations were performed using the Pearson correlation analysis.

To evaluate HBOT's effect, a mixed-model repeated-measure ANOVA model was used to compare post-treatment and pre-treatment data. The model included time, group and the group-by-time interaction. Non-imaging data analysis was followed by the Bonferroni post-hoc correction.

Brain imaging maps were analyzed using a voxel-based method to generate statistical parametric maps. Group parametric maps were corrected using the Benjamini–Hochberg False Discovery Rate (FDR) method [23]. A mixed design repeated measure ANOVA model was used to test the main interaction effect between time and group implemented in SPM

software (version 12, UCL, London, UK). A sequential Hochberg correction [24] was used to correct for multiple comparisons (P < 0.05).

Data was analyzed using SPSS software (version 22.0), and the Matlab R2019b (Mathworks, Natick, MA) Statistics Toolbox.

## Results

Between March 2018 and April 2019, 50 subjects were recruited, and 15 who did not fit the study criteria were excluded. Accordingly, 35 subjects were randomized to the HBOT (N = 18) or control (N = 17) groups. As detailed in Fig 1, one patient allocated to HBOT was not able to cooperate with the treatment protocol and preferred to stop the treatment after 20 sessions, and three patients had frequent treatment stoppages because of upper respiratory tract infections (could not equilibrate the ear pressure). Two patients from the control group refused to attend the scheduled meetings and the final analysis. Therefore, of the 35 test subjects, 14 completed the HBOT protocol and 15 completed the control protocol.

Baseline patient characteristics are summarized in Table 1. The mean age at baseline was 39.3 ± 8.1 and 32.4 ± 9.2 and the mean time from last combat exposure was 11.5 ± 5.8 and 10.3 ± 6.7 years for the HBOT and control groups, respectively. The baseline global cognitive score was on the normal range expected for the patients' age and gender, 99.4±6.2 and 98.5 ±8.7 in the HBOT and control group respectively, p = 0.75.

### Primary endpoint

Analysis of the CAPS score are summarized in Table 2. At baseline, there were no differences between the groups in any of the CAPS score parameters. A significant improvement in total CAPS score by 17.7 points (CI 11.3–24.1), with group by time interaction (F = 30.57, p<0.0001, Net effect size = 1.643, Supporting information), was demonstrated in the HBOT group. Additionally, the HBOT group had significant improvements in all of the subcategories of the CAPS score (Table 2) (Fig 2). No differences in total CAPS scores or in any of the subcategories were seen in the control group.

### Secondary endpoints

Questionnaire results are summarized in Table 3. At baseline, there were no significant differences in all questionnaire domains. Significant group-by-time interactions (F = 5.72,

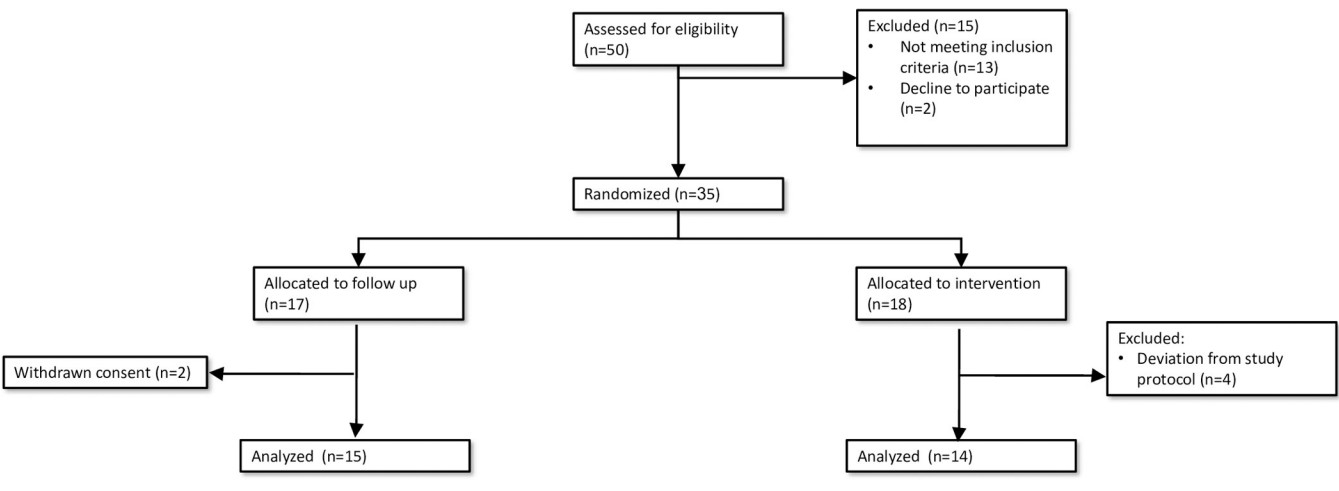

**Fig 1. Study flowchart.**

**Table 1. Patient characteristics.**

| | Treatment Group | Control Group | *P*-value |
|---|---|---|---|
| **N** | 14 | 15 | |
| **Age (y)** | 39.3±8.1 | 32.4±9.2 | 0.084 |
| **Military exposure (y)** | 6.8±3.6 | 5.2±4.8 | 0.33 |
| **Time from last combat exposure (y)** | 11.5±6.1 | 11.1±6.4 | 0.85 |
| **Total CAPS score** | 46.6±11.5 | 49.5±10.7 | 0.50 |
| **Mild PTSD (20–39)** | 3(21%) | 2(13%) | 0.65 |
| **Moderate PTSD (40–59)** | 10(71%) | 11(73%) | 1.00 |
| **Severe PTSD (60–79)** | 1(7%) | 2(13%) | 0.97 |
| **Education (y)** | 14.2±2.2 | 13.7±2.5 | 0.58 |
| **Life partner** | 9(64%) | 6(40%) | 0.27 |
| **Working** | 6(43%) | 8(53%) | 0.71 |
| **Global cognitive score***  | 99.4±6.2 | 98.5±8.7 | 0.75 |
| **Current major depression***  | 10(71%) | 13(86%) | 0.39 |
| **History pharmacotherapy** | 13(93%) | 12(80%) | 0.59 |
| **History of psychotherapy** | | | |
| **PE** | 5(36%) | 4(27%) | 0.69 |
| **EMDR** | 10(71%) | 9(60%) | 0.69 |
| **CBT** | 14(100%) | 13(87%) | 0.48 |
| **Current medications** | | | |
| **SSRI/SNRI** | 8(57%) | 8(53%) | 1.00 |
| **BDZ** | 6(43%) | 6(40%) | 1.00 |
| **Anti-psychotic** | 4(29%) | 4(27%) | 1.00 |
| **Cannabis** | 12(86%) | 10(67%) | 0.39 |
| **Cannabis (g/ month)** | 31.4±19.1 | 25.0±20.8 | 0.39 |

* normalized scores presented on an IQ-style scale, where 100 is the mean normalized score and one standard deviation of 15 points.

**Table 2. CAPS measures results.**

| | HBOT ARM (N = 14) | | | | CONTROL ARM (N = 15) | | | | Baseline | Change | Cohen's d* | ANOVA (Group-by-Time) Interaction | |
|---|---|---|---|---|---|---|---|---|---|---|---|---|---|
| | Baseline | Post HBOT | Change [95% CI] | 3 Months P value | Baseline | Control | Change [95% CI] | 3 Months P value | | | | F | P |
| B. Intrusion symptoms | 12.2±3.8 | 6.6±4.7 | -5.6 [-7.7, -3.6] | **0.000** | 12.9±2.6 | 13.1±2.1 | 0.3 [-1, 1.5] | 0.658 | 0.610 | **0.000** | 1.741 | 28.9 | **0.000** |
| C. Avoidance symptoms | 4.5±1.7 | 2.3±1.8 | -2.2 [2.9, -1.5] | **0.000** | 4.5±1.8 | 5.0±1.4 | 0.5 [-0.5, 1.4] | 0.313 | 0.960 | **0.000** | 1.797 | 23.4 | **0.000** |
| D. Cognitions and mood symptoms | 17.5±3.7 | 11.1±7.4 | -6.4 [-10.2, -2.6] | **0.003** | 16.8±5.6 | 17.5±3.8 | 0.7 [-1.2, 2.6] | 0.465 | 0.700 | **0.001** | 1.109 | 13.3 | **0.001** |
| E. Arousal and reactivity symptoms | 12.3±4.5 | 9.0±5.6 | -3.4 [-6.1, -0.6] | 0.022 | 15.3±3.2 | 15.9±3.6 | 0.7 [-0.6, 1.9] | 0.265 | 0.060 | **0.008** | 0.865 | 8.5 | **0.007** |
| T. Total | 46.6±11.5 | 28.5±17.4 | -18.1 [-25.4, -10.8] | **0.000** | 49.5±10.7 | 51.5±8.4 | 2.0 [-1.3, 5.3] | 0.211 | 0.500 | **0.000** | 1.643 | 30.6 | **0.000** |

Data are presented as mean ± SD; CI, confidence interval; Bold, significant after Bonferroni correction; * Cohen's d net effect size.

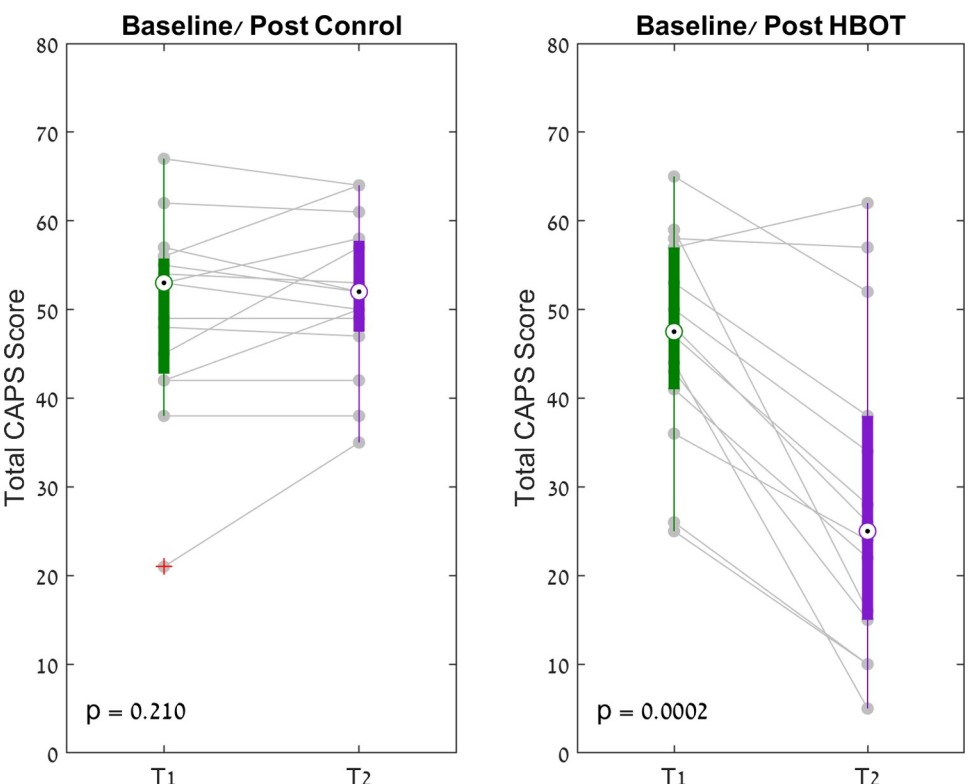

**Fig 2. CAPS scores paired box plot.** The central mark indicates the median, and the bottom and top edges of the box indicate the 25th and 75th percentiles, respectively. + Symbols indicate outliers.

**Table 3. Questionnaire results.**

| | HBOT ARM (N = 14) | | | | CONTROL ARM (N = 15) | | | | Baseline | Change | Cohen's d* | ANOVA (Group-by-Time) Interaction | |
| --- | --- | --- | --- | --- | --- | --- | --- | --- | --- | --- | --- | --- | --- |
| | Baseline | Post HBOT | Change [95% CI] | 3 Months P value | Baseline | Control | Change [95% CI] | 3 Months P value | | | | F | P |
| **BSI** | | | | | | | | | | | | | |
| Total | 38.0 ± 13.0 | 27.0 ± 16.0 | -11.0 [-19.2, -2.8] | **0.012** | 44.3 ± 7.9 | 43.8 ± 10.9 | -0.5 [-5.5, 4.6] | 0.846 | 0.134 | 0.024 | 0.890 | 5.7 | 0.020 |
| Somatization | 10.0 ± 5.5 | 7.6 ± 5.4 | -2.3 [-5.2, 0.4] | 0.092 | 12.8 ± 3.4 | 12.4 ± 5.0 | -0.4 [-2.9, 2.1] | 0.739 | 0.120 | 0.272 | 0.420 | 1.3 | 0.270 |
| Anxiety | 14.3 ± 5.1 | 10.2 ± 6.4 | -4.1 [-7.6, -0.5] | 0.027 | 17.5 ± 3.4 | 16.9 ± 3.6 | -0.6 [-2.8, 1.6] | 0.565 | 0.062 | 0.079 | 0.680 | 3.3 | 0.080 |
| Depression | 13.7 ± 4.8 | 9.1 ± 5.5 | -4.6 [-7.3, -1.8] | **0.003** | 14.0 ± 3.0 | 14.5 ± 3.8 | 0.5 [-1.4, 2.5] | 0.571 | 0.854 | **0.003** | 1.220 | 10.7 | **0.003** |
| **BECK** | 24.4 ± 6.8 | 18.1 ± 9.5 | -6.3 [-9.7, -2.9] | **0.002** | 26.8 ± 6.7 | 27.5 ± 7.7 | 0.6 [-3.5, 4.7] | 0.757 | 0.357 | **0.010** | 1.030 | 7.7 | **0.010** |

Data are presented as mean ± SD; CI, confidence interval; Bold, significant after Bonferroni correction; * Cohen's d net effect size.

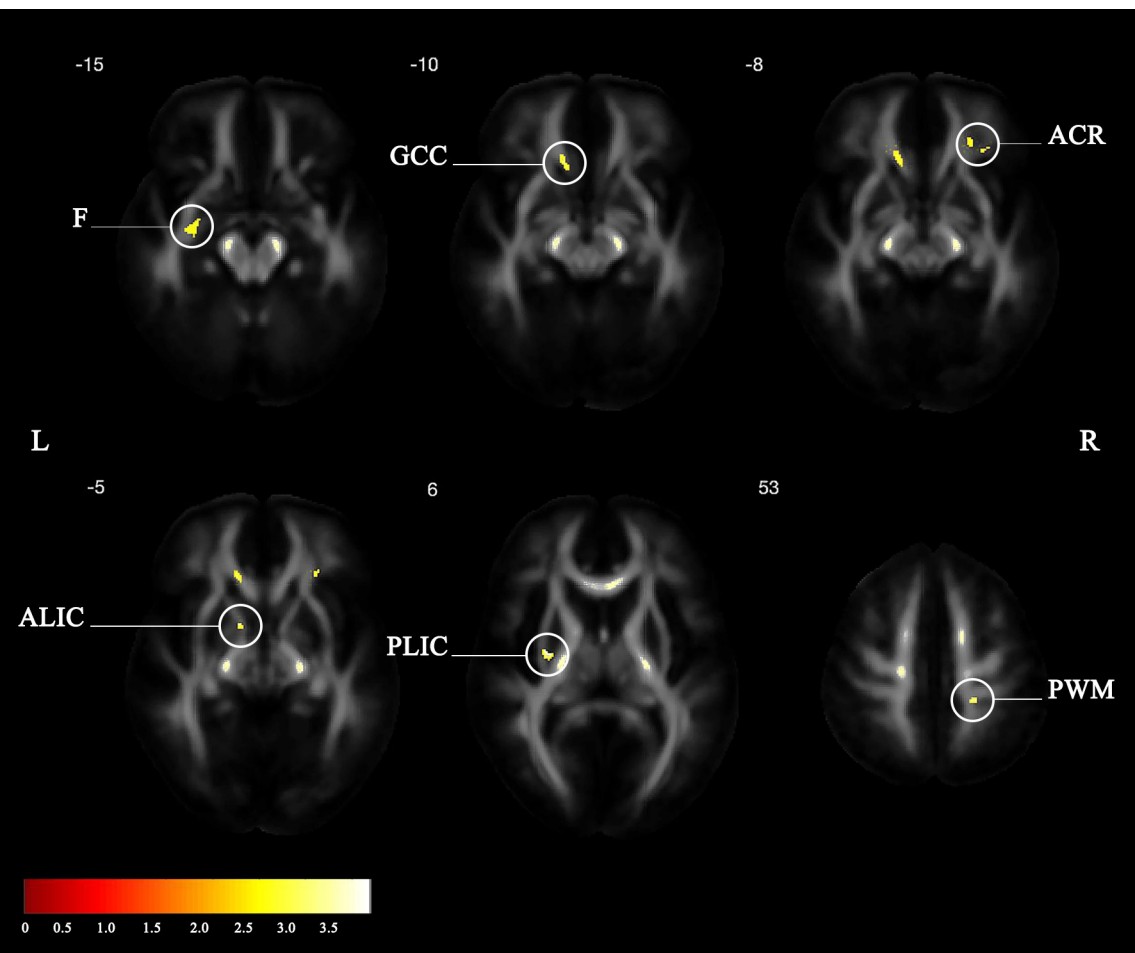

**Fig 3. Statistical parametric maps of the group-by-time interaction for the white matter FA.** (F: Fornix; GCC: Genu of corpus callosum; ACR: Anterior corona radiata; ALIC: Anterior limb of internal capsule; PLIC: Posterior limb of internal capsule; PWM: Parietal white matter).

$P$ = 0.024, Net effect size = 0.89) were demonstrated in total BSI-II scores and in the depression subcategory (F = 10.72, $P$ = 0.003, Net effect size = 1.22, S2 Table). A trend towards improvement was demonstrated in the somatization and anxiety subcategories, but the improvements did not reach statistical significance (F = 1.26, $P$ = 0.27 and F = 3.34 and $P$ = 0.079 for somatization and anxiety respectively). In addition, a significant group-time interaction (F = 7.65, $P$ = 0.01, Net effect size = 1.03, Supporting Information) was demonstrated in the total BDI-II score after HBOT. In addition, statistically significant correlations were found between the percent change in total CAPS Score and the percent change in BDI-II and BSI-18 questionnaires (r = 0.62–0.67, p<0.0004, Supporting Information).

**Regional brain microstructure integrity.** One patient did not perform MRI due to retained metal shrapnel in the lungs that was detected in a chest X-ray after inclusion. DTI-MRI was analyzed from 13 patients from the HBOT group and from 15 patients from the control group. Voxel-based DTI analysis of brain white-matter FA maps is shown in Fig 3 and in Table 4. Significant group-time interactions were demonstrated in the HBOT group compared to the control group in frontal white-matter fiber bundles connecting the thalamus and frontal lobe (anterior limb of internal capsule L and corona radiata R) and in the genu of corpus callosum, connecting between the frontal lobes. In the parietal lobe, significant clusters were found in

**Table 4. Statistical parametric maps of the group-by-time interaction for the white matter FA (L left, R right, X, sagittal, Y, coronal, Z, axial, coordinates refers to Montreal Neurological Institute.**

| Peak Region | Cluster Size | x | y | z | t Value | p Value |
|---|---|---|---|---|---|---|
| L Posterior limb of internal capsule | 79 | -29 | -8 | 6 | 3.67* | 0.0001 |
| R Parietal white matter | 93 | 21 | -34 | 53 | 3.33* | 0.001 |
| L Anterior limb of internal capsule | 46 | -11 | 5 | -5 | 3.22* | 0.001 |
| Genu of corpus callosum | 91 | 33 | 32 | -8 | 3.07 | 0.002 |
| R Anterior corona radiate | 228 | -13 | 30 | -5 | 3.04 | 0.002 |
| L Fornix | 111 | -28 | -6 | -15 | 2.8 | 0.004 |

* Satisfied Hochberg correction $p<0.05$).

parietal white-matter (adjacent to the superior longitudinal fasciculus) and in the anterior limb of the internal capsule and cerebral peduncle (ascending and descending motor and sensory fibers). Significant clusters were also found in the fornix, in an area adjacent to the hippocampus.

Only the posterior and anterior limbs of the internal capsule and parietal white-matter passed the correction to multiple comparisons ($p<0.05$, corrected). However, since this was a small sample size and we evaluate the treatment effect including a control group, we included clusters larger than 40 voxels passing p $<0.01$ uncorrected.

**Task-related functional imaging results.** Brain activity of the PTSD patients was obtained from 13 patients from the HBOT group, and 15 patients from the control group patients. The whole-brain task related activation (2-back > 0-back) at baseline and after HBOT/control sessions is shown in Fig 4 (P < 0.05, FDR corrected). The two-sample t-test analysis, performed between groups at baseline, yielded no significant functional differences. Brain clusters with a significant group-by-time interaction effect ($p<0.05$, Hochberg corrected) are listed in Table 5. Improved activity after HBOT was demonstrated in the left dorsolateral prefrontal, middle temporal and temporal gyri as well as in both thalami, left hippocampus and left insula. No significant functional differences between the two control group fMRI sessions were found (Fig 4). Statistically significant correlations were demonstrated between mean percent BOLD signal changes in peak significantly activated regions and percent change in total CAPS score (r = 0.42–0.67, p<0.05, Supporting Information).

## Safety and side effects

HBOT for PTSD was well tolerated with seven documented events of mild and spontaneously resolved middle ear barotrauma. Seven subjects from the HBOT group had an unexpected surfacing of new memories during the HBOT course. In all accept one participant memories surfaced gradually, during the second half of the treatment course (after 25–35 sessions of HBOT) in peaces that gathered to whole clear picture of the event. In one of the participants, the new memory appeared abruptly as flashback, following the fifth HBO session.

The recovery of the memories was usually accompanied by severe distress followed by integration of the memory and resolution of the distress. Patients who reported surfacing of new memories were interviewed and their symptoms and memories were documented.

No intentional questioning regarding memory surfacing was done, and thus memory surfacing could not be ruled out in other patients who might not reported as it.

## Discussion

The current study evaluates for the first time in a prospective controlled study, the effect of HBOT on veterans suffering from treatment resistant PTSD. HBOT induced significant

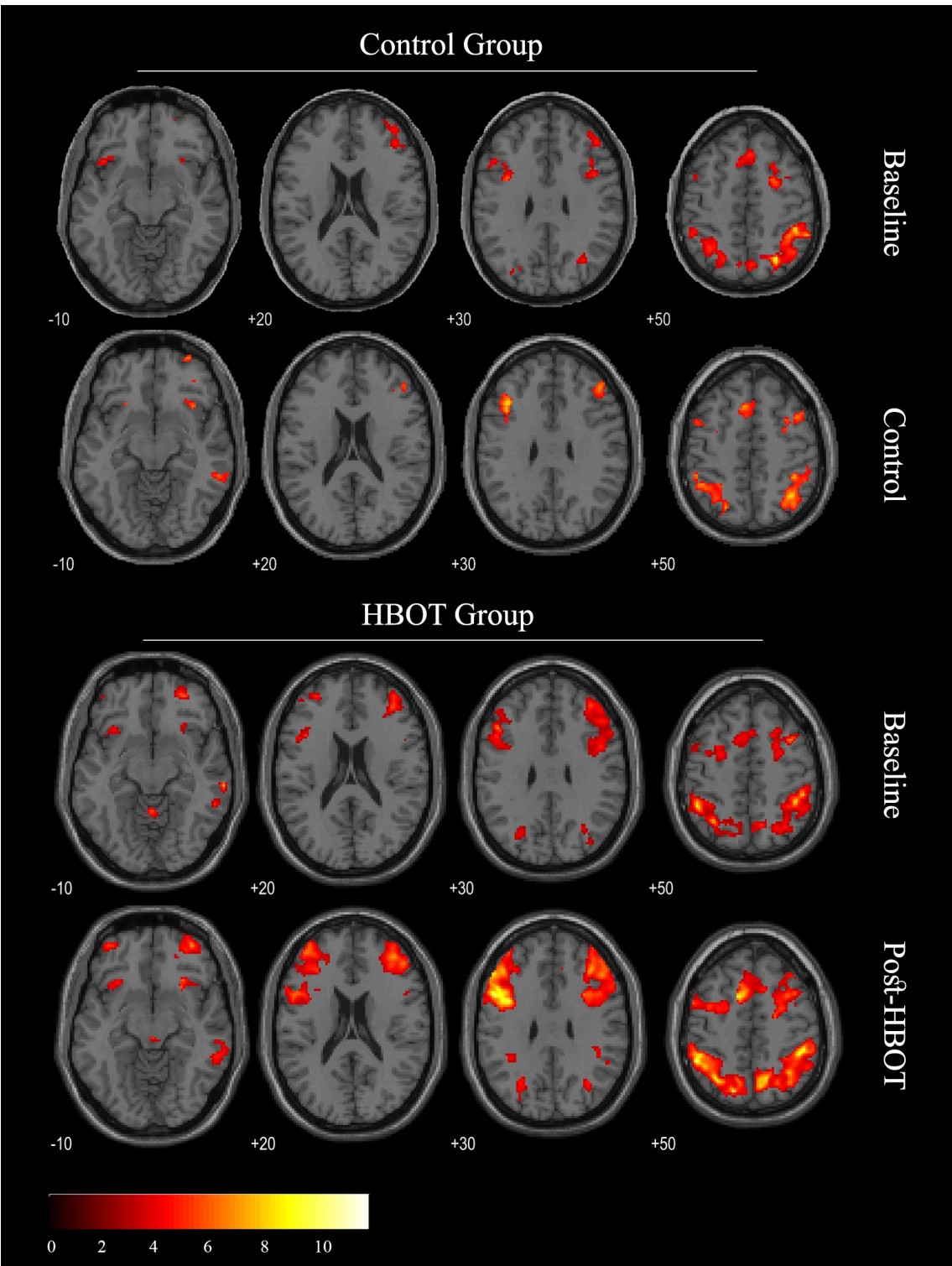

**Fig 4. Main regional loci of brain activation in a verbal working memory task (2-Back– 0-Back) Group analysis, $p < 0.05$, FDR corrected.**

**Table 5. Local maxima of brain activation (2-Back– 0-Back): Group-by-time interaction.**

| Peak Region | BA | Cluster Size | x | y | z | t value | p value |
|---|---|---|---|---|---|---|---|
| R Fusiform Gyrus | 20 | 68 | 44 | -26 | -24 | 4.85 | 0.0000 |
| L Thalamus | 50 | 78 | -12 | -10 | 8 | 4.79 | 0.0000 |
| R Thalamus | 50 | 64 | 12 | -28 | 2 | 4.67 | 0.0000 |
| L Hippocampus | 54 | 95 | -34 | -30 | -12 | 4.64 | 0.0000 |
| L Temporal Gyrus | 21 | 48 | -54 | -44 | 10 | 4.24 | 0.0001 |
| L Insula | 13 | 21 | -44 | -2 | -8 | 3.90 | 0.0001 |
| L Dorsolateral Prefrontal | 9 | 20 | -28 | 40 | 44 | 3.89 | 0.0002 |
| L Medial Posterior Parietal | 7 | 59 | -16 | -70 | 40 | 3.89 | 0.0002 |
| R PCC | 23 | 30 | 0 | -14 | 34 | 3.87 | 0.0002 |
| L Superior Temporal Gyrus | 38 | 20 | -44 | 4 | -18 | 3.82 | 0.0002 |
| L Middle Temporal Gyrus | 39 | 21 | -30 | -76 | 34 | 3.70 | 0.0003 |

reductions in the PTSD symptoms and the associated depression as assayed by CAPS-V, BDI and BSI questionnaires. The marked clinical improvement was associated with improved brain functionality and microstructural integrity as evident by fMRI and DTI-MRI.

HBOT's potential beneficial effects on combat associated PTSD was demonstrated in previous studies that evaluated its effect on TBI and included PTSD related symptoms as one of the study endpoints [14–20]. The last study from BIMA study team [20] clearly demonstrated pronounced effects of HBOT on the subgroup of veterans who had concomitant PTSD symptoms, with PTSD effected veterans benefiting more from HBOT than veterans without PTSD. Since TBI and PTSD share common symptoms such as nervousness, sleep disorders, and impaired cognitive function, it is difficult to assign the improvement to either one of the pathologies. In the current study, only patients with PTSD who did not have TBI were included, and any history of TBI served as an exclusion criterion. Thus, both clinical and radiological effects demonstrated in the current study can only be associated to HBOT's effects on PTSD.

PTSD's current treatment success rate is poor. Trauma-focused psychotherapy is currently the treatment of choice while pharmacotherapy is added when psychotherapy is insufficient. However, up to 50% fail to respond to any of the available treatments. The lack of effective treatments might have contributed to the successful recruitment in the current trial, despite the demanding treatment protocol.

Brain imaging changes can serve as markers for poor treatment responses [25]. A review of the published studies using high resolution and functional MRI techniques, indicate failure of the frontal-limbic circuit as PTSD's hallmark [5–8]. Diminished prefrontal inhibition and a hyperactive amygdala in response to both trauma-related [5,6] and non-trauma-related [6] stimuli in PTSD patients are consistent with diminished prefrontal inhibition of fear circuitry. Functional MRI studies using N-back tasks, demonstrate under-recruitment of prefrontal neurons, mostly dorsolateral PFC, and parietal cortex. Changes in brain microstructure, demonstrated by DTI-MRI, including reduced white-matter integrity, typically seen in the left frontal and temporal tracts and thalamo-cortical tracts, are also part of the fronto-limbic circuit failure [26,27]. In addition, impaired inter-hemispheric connectivity, as evident by decreased FA values in the genu of the corpus callosum, was also described among PTSD patients [28].

In the current study, using fMRI and the N-back paradigm, restoration of fronto-limbic integrity was demonstrated, with improved recruitment of the left dorsolateral PFC, of both thalami and of the left hippocampus. Improved microstructural integrity between frontal and parietal or temporal regions was also demonstrated using MRI-DTI as increased FA in the anterior limb of the left internal capsule, right corona radiata and fornix. Restoration of the

fronto-limbic circuit may explain the significant clinical improvement related to emotional regulations as reflected by a decrease in total, and in particular, criterion E of the CAPS score. Since the intrusive symptoms can also be a result of the cortex's failure to inhibit the limbic system [7], restoration of fronto-limbic circuit may also explain the significant clinical improvements in criterion B of the CAPS score.

In addition to the fronto-limbic circuit, HBOT induced significant improvements of hippocampal activity as demonstrated by fMRI and the integrity of its connections, assessed as improved FA in the fornix in DTI imaging. The hippocampus has a central role in PTSD pathogenesis, and it may serve as an important treatment target. The hippocampus is involved in memory performance and in information processing deficits observed in PTSD patients [29]. Hippocampal integrity is also crucial for fear extinction [30].

Studies on hippocampal cell culture show that HBOT can directly induce orthodromic activity and neural plasticity [31]). In addition, the dentate gyrus of the hippocampus serves as one of the major niches for endogenous neuronal stem cells (NSC), and recent reports demonstrated HBOT's effect on mitochondrial signaling and regulation of NSC proliferation and differentiation [32].

One of the interesting findings in the current study, was the surfacing of inaccessible memories in half of the patients from the HBOT group. A similar HBOT effect on childhood sexual abuse related repressed memories was previously reported in a fibromyalgia patient study [13]. It is known that direct triggering of the hippocampus by deep brain stimulation can induce surfacing of inaccessible memories [33]. Therefore, the surfacing of memories in our veteran population can be related to the direct neuroplasticity effect detailed above at the hippocampal level.

## Study limitations

First, a cohort of 35 randomized patients is rather small. Even though the results are significant, larger scale clinical trials are required to confirm the finding presented. Second, even with randomization and blinded imaging analysis, participants were not blinded to the treatment arm, due to the inherent difficulty of conducting a sham control in HBOT trials [9,2] This could possibly affect the questionnaires. However, the chronic unremitting nature of PTSD among our participants together with the correspondence between the clinical improvement and brain functional and structural improvements as evident by the brain imaging, substantiates the clinical findings. In addition, the unexpected recovery of memories and accompanied distress during the second half of the treatment course, strongly point to HBOT's direct biological effects on this cohort of PTSD patients.

**To conclude.** This prospective randomized controlled trial demonstrates that HBOT can induce neuroplasticity and improve PTSD related symptoms of veterans suffering for treatment resistant PTSD. HBOT improved both the brain function and brain microstructure in regions typically involved in PTSD pathogenesis. The correlation between the clinical improvement and the changes in the brain functionality and microstructure can shed additional important light on the biology responsible for treatment resistant PTSD.

## Supporting information

**S1 Checklist. CONSORT 2010 checklist of information to include when reporting a randomised trial**∗**.**
(DOC)

**S1 Fig. Scatter plot of the correlations between percent change in total CAPS Score and the percent change in BDI and BSI questionnaires scores.** r is Pearson's correlation coefficient, p < 0.0004 for all comparisons.
(TIF)

**S2 Fig. Scatter plot of the relative percent BOLD signal change in peak significantly activated regions, and the percent change in total CAPS score.** r is Pearson's correlation coefficient, p < 0.05 for all comparisons.
(TIF)

**S3 Fig. Questionnaire scores paired box plot.** The central mark indicates the median, and the bottom and top edges of the box indicate the 25th and 75th percentiles, respectively. + Symbols indicate outliers.
(TIF)

**S1 Table. Total CAPS score repeated measures ANOVA.**
(DOCX)

**S2 Table. Questionnaire repeated measures ANOVA.**
(DOCX)

**S1 Dataset.**
(XLSX)

**S1 File.**
(PDF)

# Acknowledgments

We are grateful to Prof. Rachel Lev-Wiesel, (The Emili Sagol University of Haifa, Israel CAT Research Center). Yair Bechor PhD, Yonathan Zemel, Katia Adler, Dr. Erez, Lang, Dr. Shachar Fynci, Dr. Nir Poliak, Dr. Gregory Fishlev, Dr. Mony Fridman, Moran Adler, Or Nagauker, Rahav Boussi-Gross, Ido May-Raz and Gil Suzin (Sagol Center for hyperbaric medicine), Merav Ben Yosef, Galina Sakliarvski (Department of radiology) and Revital Ozeri (Beit Halochem) for their participation in treatment monitoring, data acquisition and medical support. We thank Hadas Okon-Zinger (Department of Psychology, University of Haifa, Haifa, Israel) for her consultation regarding fMRI tasks. We would also like to thank all the patients for their motivation, cooperation, and confidence.

# Author Contributions

**Conceptualization:** Keren Doenyas-Barak, Shai Efrati.

**Data curation:** Merav Catalogna, Gabriela Levi.

**Formal analysis:** Merav Catalogna, Sigal Tal, Efrat Sasson.

**Investigation:** Keren Doenyas-Barak, Shir Daphna-Tekoha, Yarden Shechter.

**Methodology:** Keren Doenyas-Barak, Merav Catalogna, Amir Hadanny.

**Project administration:** Gabriela Levi, Yarden Shechter.

**Resources:** Shai Efrati.

**Software:** Merav Catalogna, Efrat Sasson.

**Supervision:** Ilan Kutz, Amir Hadanny, Shir Daphna-Tekoha, Shai Efrati.

**Validation:** Amir Hadanny, Efrat Sasson, Shai Efrati.

**Writing – original draft:** Keren Doenyas-Barak.

**Writing – review & editing:** Merav Catalogna, Ilan Kutz, Gabriela Levi, Amir Hadanny, Sigal Tal, Shir Daphna-Tekoha, Efrat Sasson, Shai Efrati.

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
