## [Decision Letter · Decision Letter 0]

5 Nov 2021

PONE-D-20-34405Hyperbaric oxygen therapy improves symptoms, brain's microstructure and functionality in veterans with treatment resistant post-traumatic stress disorder: a prospective, randomized, controlled trial.PLOS ONE

Dear Dr. Doenyas,

Thank you for submitting your manuscript to PLOS ONE. After careful consideration, we feel that it has merit but does not fully meet PLOS ONE’s publication criteria as it currently stands. Therefore, we invite you to submit a revised version of the manuscript that addresses the points raised during the review process.

I would like to sincerely apologize for the delay you have incurred with your submission. It has been exceptionally difficult to secure reviewers to evaluate your study. We have now received three completed reviews; their comments are available below. Reviewers#2 and #3 have raised significant scientific concerns about the study that need to be addressed in a revision. Please revise the manuscript to address all the reviewer's comments in a point-by-point response in order to ensure it is meeting the journal's publication criteria. Please note that the revised manuscript will need to undergo further review, we thus cannot at this point anticipate the outcome of the evaluation process.

We look forward to receiving your revised manuscript.

Kind regards,

Miquel Vall-llosera Camps

Senior Editor

PLOS ONE

Journal Requirements:

"The study was funded by the Research Fund of Shamir Medical Center, Israel.

We are grateful to Prof. Rachel Lev-Wiesel, (The Emili Sagol University of Haifa, Israel CAT Research Center). Yair Bechor PhD, Yonathan Zemel, Katia Adler, Dr.  Erez, Lang, Dr. Shachar Fynci, Dr. Nir Poliak, Dr. Gregory Fishlev, Dr. Mony Fridman, Moran Adler, Or Nagauker, Rahav  Boussi-Gross, Ido May-Raz and Gil Suzin (Sagol Center for hyperbaric medicine), Merav Ben Yosef, Galina Sakliarvski (Department of radiology) and Revital Ozeri (Beit Halochem) for their participation in treatment monitoring, data acquisition and  medical support. We thank Hadas Okon-Zinger (Department of Psychology, University of Haifa, Haifa, Israel) for her consultation regarding fMRI tasks. We would also like to thank all the patients for their motivation, cooperation, and confidence."

"The study was funded by the Research Fund of Shamir Medical Center, Israel."

Reviewers' comments:

Reviewer's Responses to Questions

**Comments to the Author**

1. Is the manuscript technically sound, and do the data support the conclusions?

Reviewer #1: Yes

Reviewer #2: Yes

Reviewer #3: Yes

2. Has the statistical analysis been performed appropriately and rigorously? 

Reviewer #1: Yes

Reviewer #2: Yes

Reviewer #3: No

3. Have the authors made all data underlying the findings in their manuscript fully available?

Reviewer #1: Yes

Reviewer #2: Yes

Reviewer #3: Yes

4. Is the manuscript presented in an intelligible fashion and written in standard English?

Reviewer #1: Yes

Reviewer #2: Yes

Reviewer #3: Yes

5. Review Comments to the Author

Reviewer #1: This is an excellent paper on a important topic. It is well written and the conclusions are sound. The groups are clearly define, as well as the diagnosis and treatment methods. It is also easy to understand. It provides a new way to address PTSD by optimizing brain function.

Reviewer #2: I thank the authors for the opportunity to review the reporting of their important research study. While I found the manuscript generally well written and concise, clarification of a couple of things could improve the quality and scientific impact of the paper in the field.

1) On page 11, it’s not clear why prescribed cannabis (mean grams/month) is reported differently than the other prescribed medication (number of patients (% of group)). The reader is left wondering what number and percentage in each group were using prescribed cannabis during the study.

2) On page 16, line 28, the authors write: "One of the interesting findings in the current study, was the surfacing of unaccusable memories in half of the patients from the HBOT group." If the adjective "unaccusable" (not charged with wrongdoing) used here was intended to convey psychological meaning regarding the nature of previously "inaccessible" memories, then "unaccusable memories" should be operationally defined here. Moreover, since this phenomenon occurred in half those receiving treatment (n=7?) and this has only been reported once previously in the literature, this could be an important psychological predicter of therapeutic response that deserves further quantitative and qualitative analysis and discussion, especially in regards to maximizing psychological interventions for PTSD. For example, quantitatively, did those who experienced these repressed memories differ on clinical outcome measures from those who didn’t experience them within the HBOT treatment group? Were any time course patterns observed in regards to when these repressed memories began to emerge following the start of treatment? For example, in the previous fibromyalgia HBOT study cited by the authors, time course information was only provided for two of the nine patients, with reemergence of repressed memories occurring between HBOT days 17 and 34 in one and HBOT days 37 and 44 in the second patient. It would also be of interest to provide more information regarding the qualitative nature of these repressed memories and how they resolved psychotherapeutically. For example, the authors should consider providing a brief and concise clinical vignette on one or more of these 7 PTSD patients as illustrative examples as was provided in the previous fibromyalgia HBOT study.

Reviewer #3: 1. Sample size determination is not acceptable. Need to add more details, e.g. effect size.

2. Data analysis section needs more work. Be consistent with the statistical terms. Just use “Repeated ANOVA” instead of “within-subject RM ANOVA ” or “mixed design RM ANOVA”. Two names were used for p value correction also: “sequence Hochberg” and “FDR (Figure 4)”, use one.

What method was used if the data does not follow normal distribution?

What is the “net effect size”?

Page 9 line 6 the sentence “univariate analyses were performed … significant variables” can be omitted. It does not seem that you performed univariable or multivariable analyses.

3. Results:

For RM ANOVA, it is not meaningful to show a significant interaction. For each outcome, better show at least three p values and mean differences (with 95% confidence intervals) for the following tests: 1) pre-post test within the treatment group, 2) pre-post test within the control group, 3) a comparison of pre-post change between the treatment and control group. The p value and mean difference (95%) for pre-post change between the two groups should be shown in Table 2 and 3.

Need to test normality for RM ANOVA.

Bonferroni not mentioned in method but not in the results.

Page 17 line 6 “affect” instead of “effect”

6. PLOS authors have the option to publish the peer review history of their article (what does this mean?). If published, this will include your full peer review and any attached files.

Reviewer #1: No

Reviewer #2: No

Reviewer #3: No

---

## [Author Response · Author response to Decision Letter 0]

17 Nov 2021

Reply to Review Comments to the Author 

Reviewer #1: 

This is an excellent paper on a important topic. It is well written and the conclusions are sound. The groups are clearly define, as well as the diagnosis and treatment methods. It is also easy to understand. It provides a new way to address PTSD by optimizing brain function.

Reply: we thank the reviewer for his support. 

Reviewer #2:

I thank the authors for the opportunity to review the reporting of their important research study. While I found the manuscript generally well written and concise, clarification of a couple of things could improve the quality and scientific impact of the paper in the field.

Comment 1: “On page 11, it’s not clear why prescribed cannabis (mean grams/month) is reported differently than the other prescribed medication (number of patients (% of group)). The reader is left wondering what number and percentage in each group were using prescribed cannabis during the study”. 

Reply: The number and percentage of patients using cannabis in each group were added to table 1.

Comment 2: “On page 16, line 28, the authors write: "One of the interesting findings in the current study, was the surfacing of unaccusable memories in half of the patients from the HBOT group." If the adjective "unaccusable" (not charged with wrongdoing) used here was intended to convey psychological meaning regarding the nature of previously "inaccessible" memories, then "unaccusable memories" should be operationally defined here.” 

Reply: Thank you for the comment. The term “unaccusable memories” was replaced by the more appropriate term “inaccessible memories”. 

Comment 3: “since this phenomenon occurred in half those receiving treatment (n=7?) and this has only been reported once previously in the literature, this could be an important psychological predicter of therapeutic response that deserves further quantitative and qualitative analysis and discussion, especially in regards to maximizing psychological interventions for PTSD. For example, quantitatively, did those who experienced these repressed memories differ on clinical outcome measures from those who didn’t experience them within the HBOT treatment group? Were any time course patterns observed in regards to when these repressed memories began to emerge following the start of treatment? For example, in the previous fibromyalgia HBOT study cited by the authors, time course information was only provided for two of the nine patients, with reemergence of repressed memories occurring between HBOT days 17 and 34 in one and HBOT days 37 and 44 in the second patient. It would also be of interest to provide more information regarding the qualitative nature of these repressed memories and how they resolved psychotherapeutically. For example, the authors should consider providing a brief and concise clinical vignette on one or more of these 7 PTSD patients as illustrative examples as was provided in the previous fibromyalgia HBOT study.”

Reply: Additional data regarding the time and pattern of the reported memories was added to the results section of the revised manuscript.

The surfacing of the repressed memories was not anticipated prior to the study so no intentional questioning and monitoring of this phenomenon was done as part of the study protocol. Since it might be possible that other participants had recovery of memories and did not report it, we cannot use the current data for additional conclusions on the psychological meaning of this effect. 

we agree that the surfacing memories has the potential to serve as a psychological predictor of therapeutic response and it indeed deserves further quantitative and qualitative analysis. Accordingly, our next study will include in depth evaluation and actively monitored of all participants with multiple aspects related to their memory. 

Reviewer #3: 

Comment 1: “Sample size determination is not acceptable. Need to add more details, e.g. effect size.

Replay: Data related to sample size calculation based on effect size was added to the Statistical analysis - Sample size section. 

Comment 2: “Data analysis section needs more work. Be consistent with the statistical terms. Just use “Repeated ANOVA” instead of “within-subject RM ANOVA ” or “mixed design RM ANOVA”. Two names were used for p value correction also: “sequence Hochberg” and “FDR (Figure 4)”, use one. 

Replay: Thank you for this comment. The ANOVA and the multiple comparisons methods were not described clearly enough. A mixed-model repeated-measure ANOVA was performed to compare post-treatment and pre-treatment data. The model included time, group and the group-by-time interaction (detailed results are presented in Supporting Tables SI-1, SI-2). 

Brain imaging parametric maps were corrected using the Benjamini–Hochberg False Discovery Rate (FDR) method (ref. 22 in the revised manuscript). Group-by-time interaction imaging analysis was corrected using the sequential Hochberg correction (ref. 23 in the revised manuscript).

The data analysis section was revised accordingly.

Comment 3: “What method was used if the data does not follow normal distribution?”

Replay: In this study, all continuous data were tested by the Kolmogorov-Smirnov test and found to be normally distributed. 

Comment 4: “What is the “net effect size”?”

Replay: Net effect size is the relative Cohen’s d effect size, defined as the improvement from baseline after HBOT minus control three months improvement divided by the pooled standard deviation (SD) of the composite score. Positive effect sizes indicate improvement. An explanation was added to the statistical analysis section.

Comment 5: “Page 9 line 6 the sentence “univariate analyses were performed … significant variables” can be omitted. It does not seem that you performed univariable or multivariable analyses.”

Replay: This was referred to the categorical data, and corrected in the revised manuscript. 

Comment 6: “Results - For RM ANOVA, it is not meaningful to show a significant interaction. For each outcome, better show at least three p values and mean differences (with 95% confidence intervals) for the following tests: 1) pre-post test within the treatment group, 2) pre-post test within the control group, 3) a comparison of pre-post change between the treatment and control group. The p value and mean difference (95%) for pre-post change between the two groups should be shown in Table 2 and 3.”

Replay: The group-by-time interaction ANOVA method is a common method for demonstrating outcomes between groups controlling for time. However, we also added p-values and mean differences with 95% confidence intervals, as suggested (Tables 2 and 3). 

Comment 7: “Need to test normality for RM ANOVA. Bonferroni not mentioned in method but not in the results.” 

Replay: Indeed, RM ANOVA model is valid for normally distributed data. All continuous data in this study were normally distributed. 

Non-imaging data was corrected for multiple comparisons using the Bonferroni correction. Marked and added to Tables 2 and 3. 

Comment 8: “Page 17 line 6 “affect” instead of “effect””

Replay: Done.

---

## [Decision Letter · Decision Letter 1]

19 Jan 2022

PONE-D-20-34405R1

Hyperbaric oxygen therapy improves symptoms, brain's microstructure and functionality in veterans with treatment resistant post-traumatic stress disorder: a prospective, randomized, controlled trial.

PLOS ONE

Dear Dr. Doenyas,

Thank you for submitting your manuscript to PLOS ONE. After careful consideration, we feel that it has merit but does not fully meet PLOS ONE’s publication criteria as it currently stands. Therefore, we invite you to submit a revised version of the manuscript that addresses the points raised during the review process.

We look forward to receiving your revised manuscript.

Kind regards,

Burak Yulug

Academic Editor

PLOS ONE

Journal Requirements:

Additional Editor Comments (if provided):

This is an interesting paper indicating the therapeutic role of HBOT on PTSD associated with relevant changes in the brain network and microstructure. Although the authors responded well to the reviewers' recommendation, I think that more detail regarding the cognitive baseline data of the patients should be included in the main text. For instance, were all patients cognitively homogeneously distributed in this respect, which could be critical in affecting both the clinical and imaging data. More relevant data regarding the questions should be provided in the main text since cognitive status could be game-changer in such clinical neuroimaging studies: What is the dementia/cognitive impairment threshold of MindStream. Why the authors did not use MMSE which is widely used for dementia exclusion.

Reviewers' comments:

Reviewer's Responses to Questions

**Comments to the Author**

1. If the authors have adequately addressed your comments raised in a previous round of review and you feel that this manuscript is now acceptable for publication, you may indicate that here to bypass the “Comments to the Author” section, enter your conflict of interest statement in the “Confidential to Editor” section, and submit your "Accept" recommendation.

Reviewer #3: All comments have been addressed

2. Is the manuscript technically sound, and do the data support the conclusions?

Reviewer #3: (No Response)

3. Has the statistical analysis been performed appropriately and rigorously? 

Reviewer #3: (No Response)

4. Have the authors made all data underlying the findings in their manuscript fully available?

Reviewer #3: (No Response)

5. Is the manuscript presented in an intelligible fashion and written in standard English?

Reviewer #3: (No Response)

6. Review Comments to the Author

Reviewer #3: (No Response)

7. PLOS authors have the option to publish the peer review history of their article (what does this mean?). If published, this will include your full peer review and any attached files.

Reviewer #3: No

---

## [Author Response · Author response to Decision Letter 1]

27 Jan 2022

Dear Prof. Burak Yulug

We thank you for your review of our manuscript. 

The Neurotrax tool box was chosen for baseline cognitive evaluation, as it is fully computerized test that was validated also among cognitively healthy subjects (1-2). The cognitive scores are presented as normalized scores according to age and education groups, on an IQ-style scale, where 100 represents the mean normalized score and one standard deviation equals to 15 points. 

The baseline global cognitive score was on the normal range expected for the patients’ age and gender, 99.4±6.2 and 98.5±8.7 in the HBOT and control group respectively, p=0.75. 

We have added the requested data regarding baseline cognitive score to the revised manuscript. 

Best regards, 

Keren Doenyas-Barak

---

## [Editor Report · Decision Letter 2]

7 Feb 2022

Hyperbaric oxygen therapy improves symptoms, brain's microstructure and functionality in veterans with treatment resistant post-traumatic stress disorder: a prospective, randomized, controlled trial.

PONE-D-20-34405R2

Dear Dr. Doenyas

We’re pleased to inform you that your manuscript has been judged scientifically suitable for publication and will be formally accepted for publication once it meets all outstanding technical requirements.

Kind regards,

Burak Yulug

Academic Editor

PLOS ONE
---

## [Editor Report · Acceptance letter]

10 Feb 2022

PONE-D-20-34405R2 

Hyperbaric oxygen therapy improves symptoms, brain's microstructure and functionality in veterans with treatment resistant post-traumatic stress disorder: a prospective, randomized, controlled trial. 

Dear Dr. Doenyas-Barak:

I'm pleased to inform you that your manuscript has been deemed suitable for publication in PLOS ONE. Congratulations! Your manuscript is now with our production department. 

Kind regards, 

on behalf of

Dr. Burak Yulug 

Academic Editor

PLOS ONE